# N-Doped Carbon/CeO_2_ Composite as a Biomimetic Catalyst for Antibacterial Application

**DOI:** 10.3390/ijms24032445

**Published:** 2023-01-26

**Authors:** Nan Wang, Xiaofan Zhai, Fang Guan, Ruiyong Zhang, Baorong Hou, Jizhou Duan

**Affiliations:** 1Key Laboratory of Marine Environmental Corrosion and Bio-Fouling, Institute of Oceanology, Chinese Academy of Sciences, 7 Nanhai Road, Qingdao 266071, China; 2Open Studio for Marine Corrosion and Protection, Pilot National Laboratory for Marine Science and Technology (Qingdao), 1 Wenhai Road, Qingdao 266237, China; 3Center for Ocean Mega-Science, Chinese Academy of Sciences, 7 Nanhai Road, Qingdao 266071, China

**Keywords:** N-C/CeO_2_ composite, catalytic materials, biomimetic catalyst, haloperoxidase-like enzyme, antibacterial

## Abstract

Exploring new and high efficiency mimic enzymes is a vital and novel strategy for antibacterial application. Haloperoxidase-like enzymes have attracted wide attention thanks to their amazing catalytic property for hypohalous acid generation from hydrogen peroxide and halides. However, few materials have displayed halogenating catalytic performance until now. Herein, we synthesized N-doped C/CeO_2_ (N-C/CeO_2_) composite materials by a combination of the liquid and solid-state method. N-C/CeO_2_ can possess haloperoxidase-like catalytic activity by catalyzing the bromination of organic signaling compounds (phenol red) with H_2_O_2_ at a wide range of temperatures (20 °C to 55 °C), with a solution color changing from yellow to blue. Meanwhile, it exhibits high catalytic stability/recyclability in the catalytic reaction. The synthesized N-C/CeO_2_ composite can effectively catalyze the oxidation of Br^−^ with H_2_O_2_ to produce HBrO without the presence of phenol red. The produced HBrO can resist typical marine bacteria like *Pseudomonas aeruginosa*. This study provides an efficient biomimetic haloperoxidase and a novel sustainable method for antibacterial application.

## 1. Introduction

Microbiological contamination constitutes one of the fatal worldwide issues facing both environmental sustainability and public healthcare [1,2]. Various antibacterial methods have been developed to limit microbial growth, such as the addition of excess chlorine dioxide [3] or antibiotics [4,5], even new types of antibacterial materials, such as nano silver [6,7]. However, the toxicity of disinfection byproducts and the bacterial resistance lead to a quest for novel and effective methods [8,9]. Therefore, it is necessary to exploit novel and effective environment-friendly and nontoxic antibacterial materials and antibacterial technology. Different conventional antibacterial agents and biomimetic catalyst emulate nature enzymes to produce intermediates such as halogenated metabolite, which target specific bacterial signaling and regulatory systems for preventing bacterial colonization or biofilm development [10].

In nature, some marine algae can effectively prevent the attachment of microorganisms by self-secreting a haloperoxidase [11,12]. This kind of haloperoxidase can catalyze the oxidation of halides (Cl^−^, Br^−^, I^−^) with H_2_O_2_ to the corresponding hypohalous acid [13]. Inspired by this phenomenon, natural haloperoxidase and functional recombinant haloperoxidase, especially vanadium haloperoxidases (V-HPOs), are applied to paint as an additive, which can effectively prevent the growth and attachment of microorganisms [14,15]. However, these natural and functional recombinant enzymes were restricted to large-scale application because of their high production costs, short-term stability, and specific reaction conditions (pH and temperature) [16]. Exploring high-performance artificial V-HPOs mimic enzymes is a useful strategy to replicate natural enzymes.

Attempts to mimic haloperoxidases reactions with synthetic enzymes have been successful in catalytic activity [17]. Research on functional enzyme mimics has seen an upsurge in recent years [10]. Several vanadium complexes [18,19] or V_2_O_5_ nanoparticles [16] have been reported as mimicking V-HPOs, which display catalytic efficiency and selectivity in oxidative halogenation reactions [20,21]. However, the vanadium compounds are mutagenic, carcinogenic, and teratogenic [17]. It is an urgent need to develop efficient and non-toxic materials to replace vanadium-based complexes. Inspired by catalyzing oxidation/halogenation reaction of cerium oxide in organic synthesis, cerium-based materials were reported to have haloperoxidase-like activity [22,23,24,25,26]. For example, cerium oxide nanorods as haloperoxidase mimic have been used in antimicrobial membranes [23,24,25]. Cerium oxides present good catalytic performance, which is attributed to the self-structural properties and environmental compatibility [25]. However, the extreme low abundance of rare-earth metallic cerium on Earth limits its large-scale application. Doping is an efficient strategy to reduce the usage amount of cerium and increase the utilization of cerium. In our previous report, compared with cerium oxide, the same amount carbon-doped cerium oxide exhibited better haloperoxidase mimicry for antimicrobial [26]. Therefore, specific doping and complex can effectively reduce resource utilization and facilitate the widespread application of cerium-based materials.

Herein, N-C/CeO_2_ composite was prepared and studied as haloperoxidase mimicry for antibacterial, as shown in Figure 1. N-C/CeO_2_ composite as a biomimetic catalyst possesses haloperoxidase-like catalytic activity by catalyzing the bromination of phenol red in the presence of H_2_O_2_ with a solution color changing from yellow to blue. Meanwhile, it can possess antibacterial application by catalyzing the oxidation of Br^−^ with H_2_O_2_ (without phenol red) to produce HBrO. The haloperoxidase activity of prepared material and the factors affecting the mimicry activity, such as temperature and concentration of catalysts, were studied. The kinetics of the catalytic reaction were investigated by varying the concentration of one reactant while keeping the concentration of others constant. Consequently, the stability and recyclability of N-C/CeO_2_ composite were proved through the reutilization test. It can effectively catalyze the oxidative bromination of Br^−^ and H_2_O_2_ to produce HBrO. The produced HBrO with a strong antibacterial activity was used to resist microorganisms such as *Escherichia coli* (*E. coli*), *Pseudomonas aeruginosa* (*P. aeruginosa*)*,* and *Staphylococcus aureus* (*S. aureus*). The proposed high efficiency N-C/CeO_2_ artificial enzyme mimic may represent a novel strategy to emulate a natural defense system for restraining biofilm growth and bacterial colonization.

## 2. Results and Discussion

### 2.1. Characterization

The morphology and crystal structure of the synthesized composites were investigated by SEM, TEM, and XRD. Figure 1A is the SEM image of the prepared N-C/CeO_2_ composite. N-C/CeO_2_ composite presents a kind of sheet morphology, and the sheets branch off from each other. The high-resolution TEM (HRTEM) image (Figure 1B) of N-C/CeO_2_ shows clear lattice fringes with an interlayer distance of 0.32 nm, which can be indexed to the (002) plane of CeO_2_. As shown in Figure 1C, the XRD pattern of N-C/CeO_2_ composite shows typical peaks at around 28°, 33°, 47°, and 56°, corresponding to the (111), (200), (220), and (311) planes of the cubic CeO_2_ (PDF#34-0394), respectively. The inset of Figure 1C displays the crystal structure illustration of CeO_2_ with cubic space group (Fm3m). The diffraction peaks are strong and sharp, implying that the N-C/CeO_2_ sample maintains good crystallinity. In addition, XRD patterns of the other N-C/CeO_2_ composites are shown in Appendix A. These results suggest that all of the XRD patterns of N-C/CeO_2_ composites are nearly identical to the pure CeO_2_. Moreover, the loading of N-doped carbon did not change the crystalline phase of the composites. As displayed in Figure 1D, the STEM elemental mapping images reveal a uniform distribution of N, C, O, and Ce. This indicates the high homogeneity of the synthesized N-C/CeO_2_ composite. To summarize, the doping modification in the current study has not changed the morphology and crystal structure of the N-C/CeO_2_ composites.

XPS was further employed to elucidate the electronic structure and chemical state of N-C/CeO_2_. As shown in Figure 1E, the refined Ce 3d XPS spectrum is composed of multiple couples of peaks, corresponding to a mixture of Ce^3+^ and Ce^4+^ oxidation states. Generally, cerium switches reversibly between its Ce(Ⅲ) and Ce(Ⅳ) states owing to the non-stoichiometric nature and multiple d-splitting of Ce element [27]. The Ce 3d XPS peaks located at 885/904 eV can be assigned to Ce^3+^ states, while the peaks at 883/901 eV, 889/908 eV, and 898/916 eV are attributed to Ce^4+^ states [28]. According to the integrated area, the N-C/CeO_2_ composite consisted of a major amount of Ce(Ⅳ) oxide (85.93%) and small amount of Ce(Ⅲ) oxide (14.08%), which is related to the lattice strain induced by Ce^3+^ and the presence of oxygen vacancies [29]. The O 1s XPS spectrum is given in Appendix A. The O1s XPS spectrum displays typical peaks at 529.5 and 531.6 eV, which can be assigned to the Ce^4+^-O and Ce^3+^-O bond, respectively. After high-temperature calcination, the final N-C/CeO_2_ product also contains abundant nitrogen-doped carbon components. The N 1s XPS spectrum (Figure 1F) of the N-C/CeO_2_ composite can be de-convoluted into three species, corresponding to pyridinic N, pyrrolic N, and oxidized N, respectively [30]. Additionally, the C 1s XPS spectrum (Figure 1G) exhibits significant signals at 284.5, 285, and 286 eV, which can be identified as C-C/C=O, C-N, and O-C=O/N-C=O functional groups, respectively [31]. In Raman spectra (Appendix A), the pure CeO_2_ delivers typical peaks at 460 cm^−1^ and 600/1170 cm^−1^, which can be assigned to the F_2g_ vibration model of the CeO_2_ and oxygen defects, respectively [32]. However, the N-C/CeO_2_ displays additional Raman spectrum peaks at around 1680 and 2900 cm^−1^, corresponding to the C-N band in the composite [33]. Meanwhile, N-C does not change the crystalline phase of the composites, which is consistent with the results of XRD. The above results indicate that the N-C/CeO_2_ composite contains ceric oxide and abundant N (O and C-)-functional groups. These are crucial to the high homogeneity of the prepared composites.

### 2.2. Haloperoxidase Mimicry Activity

To study the haloperoxidase mimicry activity of N-C/CeO_2_ composites, phenol red (PR) was used as the color substrate. N-C/CeO_2_ composite, as the haloperoxidase mimicry, can catalyze the bromination and PR in the presence of H_2_O_2_ with a solution color change from yellow to blue. N-C/CeO_2_ composite (6:1) exhibits the best haloperoxidase mimicry activity out of all of the N-C/CeO_2_ composites in Appendix A. Thus, this doping composite was used throughout the study. As shown in Figure 2A, the absorbance spectra of the solution were obtained in different reaction systems. No obvious absorption signals are detected in the c blank system with the components of NH_4_Br + H_2_O + H_2_O_2_ + c. However, there is an obvious peak at ~430 nm in systems a, b, and d, which belongs to the absorption of PR. System e reveals a distinct absorbance at 590 nm, which is attributed to the product bromophenol blue. The N-C/CeO_2_ composite can catalyze Br^−^ and PR in the presence of H_2_O_2_ to produce bromophenol blue. The corresponding color changes in different systems are shown in Figure 2A (insert). The solution is colorless in system c and yellow in systems a, b, and d. This yellow color comes from the color of dilute PR dye. The solution is blue only in system e. It indicates that the N-C/CeO_2_ composite can catalyze Br^−^ and PR in the presence of H_2_O_2_ to produce a blue-color reaction and exhibit good haloperoxidase-like catalytic activity. In addition, to further test the haloperoxidase-like activity of N-C/CeO_2_ composites, the catalytic activities of N-C composites and pure CeO_2_ were investigated as controls in Figure 2B. There is a faint peak at 590 nm of the N-C composite, which shows that the N-C composite has a certain catalytic activity. N-C/CeO_2_ delivers a significant absorbance peak at 590 nm, and its adsorption peak is higher absorbance than that of pure CeO_2_, indicating the N dopants carbon as an electron donor atom can facilitate the catalytic activity of CeO_2_ to produce HBrO [10]. These above results indicate that the N-C/CeO_2_ composite possesses excellent haloperoxidase-like activity higher than that of N-C composites and pure CeO_2_. Therefore, the use of nitrogen-doped carbon as a substrate is beneficial to improve the haloperoxidase-like catalytic activity of the N-C/CeO_2_ composite.

In order to study the haloperoxidase-like properties of N-C/CeO_2_ composite, the UV–Vis spectra of aqueous reaction were measured every 2 min within a total testing time of 40 min. As shown in Figure 2C, the absorbance at 590 nm increases quickly at the early time, while the increase of absorbance at 590 nm slows down after 30 min and tends to be stable at 40 min. The formation rate of bromination product (Br_4_PR) was evaluated by the accurate absorbance intensity of 590 nm, as displayed in Appendix A. These results show that the N-C/CeO_2_ composite has the same haloperoxidase-like catalytic activity as natural enzymes [34]. In general, the catalytic activity of artificial mimicry is associated with the working temperature. The optimal temperature of the N-C/CeO_2_ composite was investigated from 20 °C to 55 °C. The result shown in Figure 2D indicates that the catalytic activities of the N-C/CeO_2_ composite were high at various temperatures. These are only slightly affected by the temperature. Therefore, room temperature was chosen in the following experiments.

### 2.3. Kinetics Constant Determination

The reaction kinetics of the N-C/CeO_2_ composite were further studied. The Michaelis–Menten constant of substrates (H_2_O_2_, NH_4_Br, and PR) was measured by changing the concentration of one substrate, while keeping all other concentrations constant. Then, kinetics graphs (Figure 3) were obtained by the calculated initial velocity rates using kinetic data according to Equations (2) and (3). For the N-C/CeO_2_ composite, Figure 3A (left) shows the kinetic function as the concentration of the N-C/CeO_2_ composite, when the concentrations of other substrates are fixed. The kinetic values are fitted nonlinearly (blue line) according to the Michaelis–Menten equation, and the partial kinetic values are fitted linearly (dark dashed line). The results show that the kinetic data fit perfectly with the nonlinear fitting line based on the Michaelis–Menten equation. Thus, the N-C/CeO_2_ composite complies with the Michaelis–Menten kinetic of natural enzymes. The middle graph and the right graph in Figure 3A show the corresponding Lineweaver–Burk linearizations and logarithmic correlations, respectively.

The Michaelis–Menten constants (*K*_m_) and the maximal reaction rates (*v*_max_) values of all substrates (H_2_O_2_, NH_4_Br, and PR) calculated are shown in Table 1. The kinetic function, corresponding Lineweaver–Burk linearizations, and logarithmic correlations of other substrates, including H_2_O_2_, NH_4_Br, and PR, are treated with the similar method to that of the N-C/CeO_2_ substrate (Figure 3B–D). The above results show that the N-C/CeO_2_ composite as haloperoxidase mimicry matches with the catalytic reaction kinetics of natural enzymes.

The Michaelis–Menten constant of the N-C/CeO_2_ composite and vanadium bromoperoxidase (V-BPO) in the previous reports are summarized in Table 2. Compared with the V-BPO biological sample, the prepared N-C/CeO_2_ composite delivers an obviously smaller *K*_m_ of H_2_O_2_ and bigger *K*_m_ of bromide (NH_4_Br) [16,34]. Generally, *K*_m_ indicates the affinity between the substrate and catalyst. The lower *K*_m_ value of H_2_O_2_ suggests that H_2_O_2_ has a higher affinity for the surface of the N-C/CeO_2_ composite than V-BPO. The higher *K*_m_ value of Br^−^ indicates that Br^−^ has a lower affinity for the surface of the N-C/CeO_2_ composite.

To evaluate the effect of bromide source on the catalytic reaction, KBr and NaBr are used as control samples. Figure 4A shows the absorbance of the solution at 590 nm in the presence of NH_4_Br, KBr, and NaBr, respectively. The solution absorbances are almost identical for different bromide sources, meaning that the reaction is independent of the bromide source. As the stability of the catalyst is essential for real applications, reutilization tests of N-C/CeO_2_ composite are performed with the same concentration of PR, NH_4_Br, and H_2_O_2_ at room temperature. After each reaction cycle, the N-C/CeO_2_ composite is separated by centrifugation at 3020× *g* and washed with ultrapure water. Then, the obtained N-C/CeO_2_ composite is treated again with PR, NH_4_Br, and H_2_O_2_ under identical experimental conditions. As shown in Figure 4B, the absorbance at 590 nm stays almost constant through ten cycles. This clearly illustrates that the activity of the N-C/CeO_2_ composite has not decreased. The above results suggest that the catalytic activity of the N-C/CeO_2_ composite is independent of the bromide source, and it also exhibits high catalytic stability.

### 2.4. Antibacterial Test

N-C/CeO_2_ composites have good haloperoxidase mimicry activity by catalyzing the bromination of organic signaling compounds. Therefore, N-C/CeO_2_ composites can catalyze the reaction of H_2_O_2_ and Br^−^ to produce HBrO. In order to investigate the antibacterial property of N-C/CeO_2_ composites as haloperoxidase mimicry, N-C/CeO_2_ composites are applied onto the titanium plates’ surfaces and antibacterial tests are conducted. As shown in Figure 5A, the bare titanium plate/N-C/CeO_2_ composites modified titanium plates are exposed to *P. aeruginosa* suspensions at 37 °C for 4 h. Bacterial cell density and adhesion is further evaluated by fluorescence microscopy. As a control, a dense *P. aeruginosa* population is observed on the bare titanium plates surfaces in the absence of N-C/CeO_2_ composites in the medium without H_2_O_2_ and Br^−^ (Figure 5A, left column “Blank”). The same experimental set up is conducted without adding the substrates H_2_O_2_ and Br^−^ in *P. aeruginosa* suspensions. In this case, high *P. aeruginosa* adhesion/density is also observed on the N-C/CeO_2_ modified titanium plates (Figure 5A, middle column). In contrast, the absence of *P. aeruginosa* adhesion is detected on the N-C/CeO_2_ composites modified titanium plates in the presence of substrates H_2_O_2_ and Br^−^ (Figure 5A right column “N-C/CeO_2_ + Br^−^ + H_2_O_2_”). The above results indicate that the system of “N-C/CeO_2_ + Br^−^ + H_2_O_2_” exhibits the best antibacterial adhesion property. As shown in Appendix A, the proposed catalytic reaction with the prepared N-C/CeO_2_ catalyst can also work to suppress the microbial adhesion of *E. coli* and *S. aureus*. As shown in Figure 5B, the blank sample without adding N-C/CeO_2_ catalyst and Br^−^ reveals abundant *P. aeruginosa* colonies on the entire plate. Figure 5C displays the plate treated with only N-C/CeO_2_ catalyst (without Br^−^ and H_2_O_2_), and Figure 5D reveals the plate photo treated with all components of the catalytic condition (with N-C/CeO_2_, Br^−^, and H_2_O_2_). The sample with the addition of only the N-C/CeO_2_ catalyst exhibits a reduced number of *P. aeruginosa* colonies, implying that the N-C/CeO_2_ composite itself has weak antibacterial activity (Figure 5C). However, almost no *P. aeruginosa* colonies are detected on the plate of Figure 5D because of the generation of HBrO. These above results demonstrate that the N-C/CeO_2_ composite itself has weak antibacterial activity, while the N-C/CeO_2_ composites can catalyze the reaction of Br^−^ and H_2_O_2_ to produce HBrO, which plays a major role in antibacterial properties. In addition, the antibacterial properties of N-C/CeO_2_ as haloperoxidase mimicry were compared with the previously reported CeO_2_-based materials, as shown in Appendix A. N-C/CeO_2_ presents lower bacterial attachment on the titanium plates than other CeO_2_-based materials for *E. coli*, which indicates that they have good antibacterial adhesion properties. This provides a novel way to prevent biofouling and attachment to marine facilities. Therefore, N-C/CeO_2_ composites as haloperoxidase mimics have excellent bromination activity, and the produced hypobromous acid exhibits superb antibacterial activity.

## 3. Methods

### 3.1. Reagents and Apparatus

Cerium(III) nitrate hexahydrate (Ce(NO_3_)_3_•6H_2_O) was purchased from Aladdin Chemical Reagent Co., Ltd. (Shanghai, China). Phenol red (PR), melamine, ammonium bromide, NaCl, KCl, Na_2_HPO_4_, KH_2_PO_4_, acetic acid, and hydrogen peroxide solution (30%) were purchased from Sinopharm Chemical Reagent Co. Ltd. (Shanghai, China). Cell staining kit (K2081) was purchased from APExBIO (Houston, TX, USA). All the reagents and chemicals were used without further purification. All aqueous solutions were prepared with ultra-pure water (18.2 MΩ·cm) throughout this experiment. Phosphate buffered saline (PBS, 0.1 mmol·L^−1^) was prepared with 8.0 g·L^−1^ NaCl, 0.2 g·L^−1^ KCl, 1.44 g·L^−1^ Na_2_HPO_4_, and 0.44 g·L^−1^ KH_2_PO_4_ in ultra-pure water. Then, the pH of the solution is regulated to 7.0 by NaOH solution. PBS (0.1 mmol·L^−1^, 7.0) was used in the whole experiment. The aqueous standard solutions of H_2_O_2_ were stored in the dark because of their photosensitivity.

The morphology and structure investigation of the synthesized N-C/CeO_2_ composites were carried out by scanning electron microscopy (SEM, Reguas, Japan). The phase structures of these electrodeposits were determined using X-ray diffraction (XRD, Rigaku D/max-Ultima IV, Tokyo, Japan). The fine structures of these samples were further investigated by transmission electron microscopy TEM (JEM 2100F, Tokyo, Japan). The heteroatoms and functional groups were determined by X-ray photoelectron spectroscopy XPS (Escalab K-alpha 250Xi). The Raman spectra were collected on Renishaw MZ20-FC Raman microscope. The absorption spectra of UV–Vis and absorbance-time were measured with an UV–Vis spectrophotometer (UV–Vis, U-3900 HITACHI, Tokyo, Japan). Observation of bacteria was performed using a fluorescence microscopy (BX-51 with image software of Cellsens, Olympus, Japan) after staining with K2081 kit, as previously described [35].

### 3.2. Synthesis of N-C/CeO_2_ Composites

Herein, 3.0 g melamine and different amounts of Ce(NO_3_)_3_•6H_2_O were dissolved into 40 mL acetic acid and 40 mL ultra-pure water. The obtained solutions were mixed by ultrasonication for 30 min and transferred into a stainless-steel vessel. The hydrothermal reaction was carried out at 120 °C for 12 h. Thereafter, the solvent was removed from the product by the freezing drying process using vacuum equipment. The resultant materials were annealed at 520 °C for 4 h at a ramp rate of 5 °C·min^−1^ in the air. Different N-C/CeO_2_ composites were obtained by varying the mass ratio of melamine and Ce(NO_3_)_3_•6H_2_O (6:1, 3:1, 2:1, 1:1, 1:2).

### 3.3. Haloperoxidase-like Activity of N-C/CeO_2_ Composites

The haloperoxidase-like activity of the synthesized N-C/CeO_2_ composites was analyzed using an optical absorption spectroscopy. The reaction scheme was as follows: N-C/CeO_2_ composite catalyzes the oxidative bromination H_2_O_2_ and Br^−^ in the presence of PR, resulting in the color change from yellow to blue. The 950 μL mixed solutions (containing 28 μmol·L^−1^ PR, 69.4 mmol·L^−1^ NH_4_Br, 830 μmol·L^−1^ H_2_O_2_, and 50 μg·mL^−1^ N-C/CeO_2_ composites) reacted at room temperature for 40 min. Afterwards, the absorption was measured by UV–Vis spectroscopy. As a control, the absorption spectra of mixtures were measured when one of the substrates was absent in all mixtures. The amount of the added reagent was quantified. In addition, the optimal reaction conditions such as temperature, H_2_O_2_ concentration, and N-C/CeO_2_ composite concentration were tested by changing one reaction condition while leaving other conditions unchanged. Three replicate experiments were performed.

### 3.4. Determination of Kinetic Constant

The kinetic constants were carried out in time course mode of UV–Vis by fixing the wavelength at 590 nm [20]. The absorbance of mixed solutions was measured by changing the concentration of one reactant while keeping others constant in kinetic tests. In order to obtain the optimal concentration of all reactants, each measurement was carried out at 590 nm for 40 min. In addition, kinetic parameters were calculated based on the slopes (dA_590nm_/dt), which were kept constant over 5 min. The kinetic constants (Michaelis–Menten constant *K*_m_ and the maximum reaction velocity *v*_max_) were obtained using the Linewaver–Burk linearization (Equation (1)) [36,37].
1/*v* = *K*_m_/*v*_max_[C] + 1/*v*_max_(1)
where *v* is the initial velocity and C is the concentration of substrate. In order to evaluate the *K*_m_ and *v*_max_, *v* was calculated. In our experiments, the product Br_4_PR was used as a measure of the reaction rate to obtain the initial reaction rate (Equation (2)).
*v* = d[Br_4_PR]/dt(2)

The Br_4_PR concentration was obtained according to the Lambert–Beer law [25] (Equation (3)).
[Br_4_PR] = A_590_/dε_Br4PR_(3)
where ε_Br4PR_ is the extinction coefficient of Br_4_PR and its value is 72,200 L·mol^−1^·cm^−1^.

In addition, in order to test the dependence on the Br^−^ source, some Br-salts such as KBr, NaBr, and NH_4_Br served as the Br^−^ source in the mixture reaction solutions. The reutilization test of N-C/CeO_2_ composite was carried out in ten recycles with N-C/CeO_2_, PR, NH_4_Br, and H_2_O_2_.

### 3.5. Bacterial Adhesion Tests

For bacterial adhesion tests, *P. aeruginosa* (Gram-negative, typical marine bacterium, risk group 2 organism) as a model bacteria was grown in Luria Broth (LB) medium with shaking at 160 rpm and 37 °C for 12 h. The cell concentration of *P. aeruginosa* in the medium was calculated using the plate colony counting method [38]. Here, 10^7^ colony-forming units (cfu) mL^−1^  *P. aeruginosa* were separately obtained by centrifugation. These cells were resuspended in 0.1 mmol·L^−1^ PBS to obtain a cell concentration of 10^7^ cfu·mL^−1^. Multiple sets of 10 mL of this PBS cell suspension solutions were placed into 50 mL inoculation tubes and used for later bacterial adhesion tests.

Titanium plates (Ti, 0.1 × 1 × 1 cm^3^), with/without N-C/CeO_2_ composite, were placed into the above bacterial solution and cultivated with agitation at 37 °C for 4 h in different systems: (1) Ti without Br^−^ and H_2_O_2_ (blank), (2) modified Ti without Br^−^ and H_2_O_2_ (N-C/CeO_2_), and (3) modified Ti with Br^−^ and H_2_O_2_ (N-C/CeO_2_ + Br^−^ + H_2_O_2_). Three replicate experiments were performed per system. Afterwards, Ti was stained using a staining kit (K2081) for 15 min in the dark. Excess stain was gently removed by sterile PBS. The stained samples were examined by fluorescence microscopy. Bacterial solutions containing the same concentration of N-C/CeO_2_ composite, Br^−^, and H_2_O_2_ were cultured on an agar plate at 37 °C for 24 h. Three parallel agar plates were painted for each bacterial solution. As a control, PBS bacteria solutions containing the N-C/CeO_2_ composite were cultured under the same conditions. Three parallel experiments were performed. *E. coli* (Gram-negative) and *S. aureus* (Gram-positive) were also studied using the same experimental method. These above results show that the N-C/CeO_2_ composite as haloperoxidase mimicry presents good antibacterial activity.

## 4. Conclusions

In summary, the N-C/CeO_2_ composite was successfully synthesized using melamine as a carbon and nitrogen source. The N-C/CeO_2_ composite can effectively catalyze the oxidation of H_2_O_2_ with bromination of organic signaling compounds to produce a blue-color reaction, and presents excellent intrinsic haloperoxidase mimicry activity. The catalytic activity of the N-C/CeO_2_ composite is influenced by the substrate concentration and almost not influenced by temperature. The N-C/CeO_2_ composite as haloperoxidase mimicry can catalyze the reaction process of H_2_O_2_, Br^−^, and PR, which complied with the typical Michaelis–Menton kinetics process. The N-C/CeO_2_ composite shows good catalytic stability and recyclability in multiple reaction cycles. In the absence of phenol red, the produced HBrO catalyzed by N-C/CeO_2_ composites presents good antibacterial activity against the model bacteria, especially *P. aeruginosa*. The N-doped carbon/CeO_2_ composite as a biomimetic catalyst for antibacterial application is a novel and efficient “green” strategy to emulate and utilize a natural defense system for preventing bacterial colonization and biofilm growth. However, the catalytic activity of the N-doped carbon/CeO_2_ composite is mainly attributed to the action of CeO_2_, and the formation mechanism of the halogenated reactive oxygen species needs to be further improved. This work introduces a stable, green, and environment-friendly biomimetic material for antibacterial applications.

## Data Availability

The data that support the plots within this paper are available from the corresponding author upon reasonable request.

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
