# Peer review of "N-Doped Carbon/CeO2 Composite as a Biomimetic Catalyst for Antibacterial Application"

_ijms, 2023, doi:10.3390/ijms24032445_

Round 1

Reviewer 1 Report

Overall, this paper was enjoyable to read and the authors have demonstated a new stable biomimetic material for antibacterial applications. The experiments selected were logical and analysed to a high standard.   

Abstract

Clear and concise

Change “Exploring new and high efficiency mimics enzyme” to “Exploring new and high efficiency mimic enzymes.”

Introduction

Effectively highlights the problem. 

Sufficient background was provided for the reader.

List bacteria in alphabetical order (page 2)

Results and Discussion

Clear informative high resolution figures used

Well written section with very minor corrections.

Figure 3 – add what the blue and dark dashed line is (even though it is mentioned in the main text).

Include:

(1)  “g” or “rcf” units instead of 6000 rpm (page 7)

(2)   bacterial concentration in Figure 5 (page 8) - 107 cfu·mL-1

(3)  the word “agar” in “P. aeruginosa plate photo” so its not confused with the titanium plate.

Methods

Not completely reproducible. For example, 

(1)  Include pH of PBS used.

(2)  Page 9 – correct formatting - “12 h.      whereafter”. 

(3)  Section 3.3 – what volume was used for the assay? Number of replicates used?

(4)  Section 3.5: 

- Include risk group status for P. aeruginosa (Biosafety level 2 or risk group 2 organism)

-Include a reference for the plate colony counting method.

- Were 6-well, 12-well or individual 12 mL inoculation tubes used for bacterial/titanium exposure assays?

-Include number of replicates. 

-Was it 3 hr or 4 hr for exposure of metal to bacteria (reported 4 hr in results section and 3 hr in methods).

-What are you staining for? - strong green fluorescence indicates viable cells, red cells are dead. 

-What volumes of reagents were used to test the three different systems (page 12)

References

A good use of accurate references from relevant sources. 

Author Response

Overall, this paper was enjoyable to read and the authors have demonstated a new stable biomimetic material for antibacterial applications. The experiments selected were logical and analysed to a high standard.

Response: We thank the Reviewer for the positive comments and affirmation of this study. In the revised manuscript, we have carefully checked the manuscript and corrected the problems (see details below).

Abstract

Clear and concise

Change “Exploring new and high efficiency mimics enzyme” to “Exploring new and high efficiency mimic enzymes.”

Response: Thanks for your helpful comment. The corresponding description have been changed to “Exploring new and high efficiency mimic enzymes.” in Abstract.

Introduction

Effectively highlights the problem.

Sufficient background was provided for the reader.

List bacteria in alphabetical order (page 2)

Response: Thanks for your valuable comment. We have revised the order of bacteria according to the alphabetical order. The corresponding descriptions have been modified and highlighted in page 2 as follows:

“The produced HBrO with a strong antibacterial activity, was used to resist microorganisms such as Escherichia coli (E. coli), Pseudomonas aeruginosa (P. aeruginosa) and Staphylococcus aureus (S. aureus).”

Results and Discussion

Clear informative high resolution figures used

Well written section with very minor corrections.

Figure 3 – add what the blue and dark dashed line is (even though it is mentioned in the main text).

Response: Thanks for your helpful comment. We have noted the blue and dark dashed line in Figure 3 caption as follows:

“Figure 3. Kinetics of the fourfold oxidative bromination of PR as a function of the substrate concentrations, note: C represents N-C/CeO2 composite, blue line: nonlinearly fitted line of kinetic values, dark dashed line: the linearly fitted line of partial kinetic values.”

Include:

(1) “g” or “rcf” units instead of 6000 rpm (page 7)

(2) bacterial concentration in Figure 5 (page 8) -107 cfu·mL-1

(3) the word “agar” in “P. aeruginosa plate photo” so its not confused with the titanium plate.

Response: Thank you for your professional comments. We have carefully modified and highlighted the unit/writing errors in the revised manuscript

 as follows:

(1) “6000 rpm” has been changed into “3020 g” in page 7. The corresponding description is that “After each reaction cycle, the N-C/CeO2 composite is separated by centrifugation at 3020 g”.

(2) The bacterial concentration 107 cfu·mL-1 has been added into the caption of Figure 5 in page 5. The corresponding description is that “Figure 5 (A) Cell staining images of P. aeruginosa treated with titanium plate in different systems: blank, N-C/CeO2 and N-C/CeO2 + Br- + H2O2, P. aeruginosa agar photo (B) blank, (C) after adding N-C/CeO2 and (D) after adding N-C/CeO2 + Br- + H2O2. Note: the concentration of P. aeruginosa is 107 cfu·mL-1.”

(3) “P. aeruginosa plate photo” has been changed into “P. aeruginosa agar photo” in caption of Figure 5. The corresponding description is that “Figure 5 (A) Cell staining images of P. aeruginosa treated with titanium plate in different systems: blank, N-C/CeO2 and N-C/CeO2 + Br- + H2O2, P. aeruginosa agar photo (B) blank, (C) after adding N-C/CeO2 and (D) after adding N-C/CeO2 + Br- + H2O2. Note: the concentration of P. aeruginosa is 107 cfu·mL-1.”

Methods

Not completely reproducible. For example,

(1) Include pH of PBS used.

Response: Thank you for your valuable comments. We have replenished a detail description in Reagents and Apparatus in the revised manuscript. The corresponding description about pH of PBS has been added as follow:

“Phosphate buffered saline (PBS, 0.1 mmol·L-1) was prepared with 8.0 g·L-1 NaCl, 0.2 g·L-1 KCl, 1.44 g·L-1 Na2HPO4, 0.44 g·L-1 KH2PO4 in ultra-pure water. Then, the pH of the solution is regulated to 7.0 by NaOH solution. PBS (0.1 mmol·L-1, 7.0) was used in the whole experiment.”

(2) Page 9 – correct formatting - “12 h. whereafter”.

Response: Thank you for your helpful comments. “12 h.  whereafter” has been changed to “12 h. Whereafter” in the revised manuscript.

(3) Section 3.3 – what volume was used for the assay? Number of replicates used?

Response: Thank you for your professional comments. The volume used for the assay was 950 μL and three replicate experiments were performed in this work. The corresponding modifications were given and highlighted as follows in the revised manuscript:

“The 950 μL mixed solutions (containing 28 μmol·L-1 PR, 69.4 mmol·L-1 NH4Br, 830 μmol·L-1 H2O2 and 50 μg·mL-1 N-C/CeO2 composites) reacted at room temperature for 40 min.”

“Three replicate experiments were performed.”

(4) Section 3.5:

-Include risk group status for P. aeruginosa (Biosafety level 2 or risk group 2 organism)

-Include a reference for the plate colony counting method.

- Were 6-well, 12-well or individual 12 mL inoculation tubes used for bacterial/titanium exposure assays?

-Include number of replicates.

-Was it 3 hr or 4 hr for exposure of metal to bacteria (reported 4 hr in results section and 3 hr in methods).

-What are you staining for? - strong green fluorescence indicates viable cells, red cells are dead.

-What volumes of reagents were used to test the three different systems (page 12)

Response: Thank you for your valuable comments. We have revised point-to-point as follows:

Risk group of P. aeruginosa is belong to 2 according to the classification of German TRBA. The corresponding description has been changed into “P. aeruginosa (Gram-negative, typical marine bacterium, risk group 2 organism)”

The reference for the plate colony counting method has been added into revised manuscript. “The cell concentration of P. aeruginosa in the medium was calculated by plate colony counting method [38].”

  1. Kim, D. J.; Chung, S. G.; Lee, S. H.; Choi, J. W., Relation of microbial biomass to counting units for Pseudomonas aeruginosa. African Journal of Microbiology Research 2012, 6, (21), 4620-4622.

50 mL inoculation tubes were used for bacterial/titanium exposure assays. The corresponding description has been added into the revised manuscript as follow:

“Multiple sets of 10 mL this PBS cell suspension solutions were putted into 50 mL inoculation tubes and used for later bacterial adhesion tests.”

Three replicate experiments were performed in this wok. The corresponding description has been added into the revised manuscript as follows:

“Three replicate experiments were performed per system.”

“Three parallel agar plates were painted each bacterial solution.”

Metal plates were exposed bacterial for 4 h. We have been corrected in Section 3.5 of revised manuscript as follow:

“Titanium plates (Ti, 0.1*1*1 cm3), with / without N-C/CeO2 composite, were put into the above bacterial solution and cultivated with agitation at 37 oC for 4 h in different systems:”

The staining method is used to clearly observe the adhesion of bacteria and the growth state of bacteria on metal surfaces. The staining kit (K2081) can show different colors for live and dead bacteria under fluorescence microscope. That is, strong green fluorescence indicates viable cells, red cells are dead.

10 mL PBS cell suspension solutions were used to test the three different systems. The corresponding description were given in the end sentences of last paragraph in our revised manuscript. “Multiple sets of 10 mL this PBS cell suspension solutions were putted into 50 mL inoculation tubes and used for later bacterial adhesion tests.”

References

A good use of accurate references from relevant sources.

Response: Thank you for your positive comments.

Reviewer 2 Report

In this manuscript, the authors reported that the N doped C/CeO2 (N-C/CeO2) composite materials by a combination of the liquid and solid-state method. N-C/CeO2 can possess haloperoxidase-like catalytic activity by catalyzing the bromination of organic signaling compounds (phenol red) with H2O2 in a wide range of temperature with a solution color changing from yellow to blue. Meanwhile, it exhibits high catalytic stability/recyclability in the catalytic reaction. The synthesized N-C/CeO2 composite can effectively catalyze the oxidation of Br- with H2O2 to produce HBrO without the present of phenol red. The produced HBrO can resist Pseudomonas aeruginosa et al. bacteria. This study provides an efficient bio-mimetic haloperoxidase and a novel sustainable method for antibacterial application. In overall, this manuscript is interesting but in order to consider publication, this work should be revised. The following comments should be addressed for the improvement of their manuscript.

Comment 1: The overall study aims for this study about the N doped carbon/CeO2 composite as biomimetic catalyst for antibacterial application need to be further clarified in detail as compared to current conventional antibacterial application antibacterial application. The advantages and disadvantages of the novel N doped carbon/CeO2 composite as biomimetic catalyst for antibacterial application need to be clarified in detail.

Comment 2: The various recent reports and their research findings on the “N doped C/CeO2 (N-C/CeO2) composite materials” using various potential techniques in improving the antibacterial application should be summarized into a table form and discussed for better understanding in term of benchmarking points with your research findings.

Comment 3: The detailed raman analysis can be included to further provide detailed information about chemical structure, phase and molecular interactions between N doped carbon/CeO2 composite for better fundamental understanding.

Comment 4: What is the role of N dopants within the lattice of carbon/CeO2 composite or element partitioning in improving the antibacterial application performance? Please discuss and clarify with fundamental support. 

Comment 5: The carefully English correction is necessary for the whole manuscript. Please check and revise accordingly.

Author Response

In this manuscript, the authors reported that the N doped C/CeO2 (N-C/CeO2) composite materials by a combination of the liquid and solid-state method. N-C/CeO2 can possess haloperoxidase-like catalytic activity by catalyzing the bromination of organic signaling compounds (phenol red) with H2O2 in a wide range of temperature with a solution color changing from yellow to blue. Meanwhile, it exhibits high catalytic stability/recyclability in the catalytic reaction. The synthesized N-C/CeO2 composite can effectively catalyze the oxidation of Br- with H2O2 to produce HBrO without the present of phenol red. The produced HBrO can resist Pseudomonas aeruginosa et al. bacteria. This study provides an efficient bio-mimetic haloperoxidase and a novel sustainable method for antibacterial application. In overall, this manuscript is interesting but in order to consider publication, this work should be revised. The following comments should be addressed for the improvement of their manuscript.

Response: We thank the Reviewer for the positive comments. According to your helpful suggestions, we have carefully revised the manuscript. Your valuable comments are highly helpful for improving the quality of our work. In the revised manuscript, we have carefully checked the manuscript and corrected the problems (see details below).

Comment 1: The overall study aims for this study about the N doped carbon/CeO2 composite as biomimetic catalyst for antibacterial application need to be further clarified in detail as compared to current conventional antibacterial application antibacterial application. The advantages and disadvantages of the novel N doped carbon/CeO2 composite as biomimetic catalyst for antibacterial application need to be clarified in detail.

Response: Thanks for your valuable comments. We have added a brief description about the current antibacterial methods as follows in Introduction section of the revised manuscript. “Various antibacterial methods have been developed to limit microbial growth, such as the addition of excess chlorine dioxide [3] or antibiotics [4, 5], even new types antibacterial materials, nano silver [6, 7]. However, the toxicity of disinfection byproducts and the bacterial resistance leads to a quest for novel and effective method [8, 9].” In addition, the biomimetic catalyst for antibacterial application has been clarified in Introduction as follow:

“Different conventional antibacterial agents, biomimetic catalyst emulates nature enzyme to produce intermediates such as halogenated metabolite, which target specific bacterial signaling and regulatory systems for preventing bacterial colonization or biofilm development [10].” and “N-C/CeO2 composite as biomimetic catalyst, possesses haloperoxidase-like catalytic activity by catalyzing the bromination of phenol red in the presence of H2O2 with a solution color changing from yellow to blue. Meanwhile, it can possess antibacterial application by catalyzing the oxidation of Br- with H2O2 (without phenol red) to produce HBrO.”

N doped carbon/CeO2 composite as biomimetic catalyst for antibacterial application is a novel and efficient “green” strategy to emulate and utilize a natural defense system for preventing bacterial colonization and biofilm growth. However, the catalytic activity of N doped carbon/CeO2 composite is mainly attributed to the action of CeO2, and the formation mechanism of the halogenated reactive oxygen species need to be further improved. We have been added the advantages and disadvantages of the novel N doped carbon/CeO2 composite as biomimetic catalyst for antibacterial application in the Conclusion section of the revised manuscript as follow:

“N doped carbon/CeO2 composite as biomimetic catalyst for antibacterial application is a novel and efficient “green” strategy to emulate and utilize a natural defense system for preventing bacterial colonization and biofilm growth. However, the catalytic activity of N doped carbon/CeO2 composite is mainly attributed to the action of CeO2, and the formation mechanism of the halogenated reactive oxygen species need to be further improved.”

Comment 2: The various recent reports and their research findings on the “N doped C/CeO2 (N-C/CeO2) composite materials” using various potential techniques in improving the antibacterial application should be summarized into a table form and discussed for better understanding in term of benchmarking points with your research findings.

Response: Thank you for your helpful comments. We have summarized and compared the recent reported CeO2 based materials with our N-C/CeO2 in antibacterial application as shown in table s1 in electronic supporting information. It is obvious that the CeO2 based materials as haloperoxidase mimicry present different antibacterial properties for E. coli. The CeO2 based materials exhibit high bacterial attachment on the substrate of PVA fibers and agar plates. The CeO2@C and N-C/CeO2 in our work present very low bacterial attachment on the Titanium plates, which indicates they have good antibacterial adhesion properties. This provides a novel way to prevent biofouling and attachment to marine facilities. The corresponding description has been added and highlighted in the Section 2.4 of the revised manuscript as follow:

“In addition, the antibacterial properties of N-C/CeO2 as haloperoxidase mimicry were compared with the previous reported CeO2 based materials as shown in table s1. N-C/CeO2 presents pretty lower bacterial attachment on the Titanium plates than other CeO2 based materials for E. coli, which indicates they have good antibacterial adhesion properties. This provides a novel way to prevent biofouling and attachment to marine facilities.”

Table 1 The antibacterial properties of CeO2 based materials as haloperoxidase mimicry.

Catalysts

Substrate

Bacteria

Bacterial attachment (%)

Reference

CeO2−x NRs

PVA fibers

E. coli

~37

ACS Appl. Mater. Interfaces 2018, 10, 44722−44730

CeO2−x nanorods with different aspect ratios

Agar plates

E. coli

~42

ACS Sustainable Chemistry & Engineering 2020, 6744-6752

CeO2@C

Titanium plates

E. coli

~2.5

Colloids and Surfaces A: Physicochemical and Engineering Aspects

N-C/CeO2

Titanium plates

E. coli

~0.24

This work

  1. Hu, M.; Korschelt, K.; Viel, M.; Wiesmann, N.; Kappl, M.; Brieger, J.; Landfester, K.; Therien-Aubin, H.; Tremel, W., Nanozymes in nanofibrous mats with haloperoxidase-like activity to combat biofouling. ACS Applied Materials & Interfaces 2018, 10, (51), 44722-44730.
  2. He, X.; Tian, F.; Chang, J.; Bai, X.; Yuan, C.; Wang, C.; Neville, A., Haloperoxidase mimicry by CeO2–x nanorods of different aspect ratios for antibacterial performance. ACS Sustainable Chemistry & Engineering 2020, 8, (17), 6744-6752.
  3. Wang, N.; Li, W.; Ren, Y.; Duan, J.; Zhai, X.; Guan, F.; Wang, L.; Hou, B., Investigating the properties of nano core-shell CeO2@C as haloperoxidase mimicry catalyst for antifouling applications. Colloids and Surfaces A: Physicochemical and Engineering Aspects 2021, 608, 125592.

Comment 3: The detailed raman analysis can be included to further provide detailed information about chemical structure, phase and molecular interactions between N doped carbon/CeO2 composite for better fundamental understanding.

Response: Thanks for your professional comments. We have replenished the Raman spectra of the CeO2 and N-C/CeO2 materials in the revised manuscript. The Raman spectra were collected on Renishaw MZ20-FC Raman microscope as shown in Figure s3 in electronic supporting information. Raman spectra of the CeO2 and N-C/CeO2 have been recorded in the range 200~3200 cm-1. For the CeO2, an obvious peak at 460 cm-1 is assigned to the Raman active F2g vibration model of the CeO2 and two weak bands ~600 and 1170 cm-1 are related to the presence of oxygen defects [J. Phys. Chem. C, 2012, 116, 10009–10016] in the CeO2 and N-C/CeO2. Except that, there are two wide bands ~1680 and ~2900 cm-1 corresponding to the C-N band in N-C/CeO2 [Sustainable Energy Fuels,2017,1,288–298]. These results indicates that the N-C/CeO2 composite contains ceric oxide and N-C-functional groups. Meanwhile, the N-C doping does not change the crystalline phase of the CeO2, which is consistent with the results of XRD.

The figure 3 has been added into the electronic supporting information.

Figure s3 Raman spectra of CeO2 and N-C/CeO2 in the range 200~3200 cm-1.

The corresponding description has been added into the Section 2.1 and the Section 3.1 in the revised manuscript as follow:

“In Raman spectra (Figure s3), the pure CeO2 delivers typical peaks at 460 cm-1 and 600/ 1170 cm-1, which can be assigned to F2g vibration model of the CeO2 and oxygen defects, respectively [32]. Except that, the N-C/CeO2 displays additional Raman spectrum peaks at around 1680 and 2900 cm-1, corresponding to the C-N band in the composite [33]. Meanwhile, the N-C does not change the crystalline phase of the composites, which is consistent with the results of XRD.”

“The Raman spectra were collected on Renishaw MZ20-FC Raman microscope.”

Reference

  1. Du, X.; Zhang, D.; Shi, L.; Gao, R.; Zhang, J., Morphology Dependence of Catalytic Properties of Ni/CeO2 Nanostructures for Carbon Dioxide Reforming of Methane. The Journal of Physical Chemistry C 2012, 116, (18), 10009-10016.
  2. Maiti, S.; Dhawa, T.; Mallik, A. K.; Mahanty, S., CeO2@C derived from benzene carboxylate bridged metal-organic frameworks: ligand induced morphology evolution and influence on the electrochemical properties as a lithium-ion battery anode. SUSTAINABLE ENERGY & FUELS 2017, 1, (2), 288-298.

Comment 4: What is the role of N dopants within the lattice of carbon/CeO2 composite or element partitioning in improving the antibacterial application performance? Please discuss and clarify with fundamental support.

Response: Thank you for your helpful comments. The previous researches report [Adv. Mater. 2018, 1707073] that the ligands with electron donor atoms (N-function) have an impact on the accessibility and the stability of the metal oxidation state, thereby affecting the oxygen storage and product release (HBrO formation) behavior. In addition, we have conducted controlled experiments through comparison the adsorption peak of N-C, CeO2 and N-C/CeO2 as shown in Figure 2B. N-C composite exhibits negligible absorbance compared with CeO2 and N-C/CeO2. The N-C/CeO2 presents higher absorbance than pure CeO2, which indicates that N dopants carbon facilitates the catalytic activity of CeO2 to produce HBrO, thereby improving the antibacterial performance. We have been added and supplemented in Section 2.2 in the revised manuscript as follow:

“N-C/CeO2 delivers a significant absorbance peak at 590 nm, and its adsorption peak is higher absorbance than pure CeO2, indicating the N dopants carbon as electron donor atom can facilitate the catalytic activity of CeO2 to produce HBrO [10].”

Comment 5: The carefully English correction is necessary for the whole manuscript. Please check and revise accordingly.

Response: Thank you for your helpful comments. We have carefully checked and revised the English spelling and language issues of the whole manuscript. Some modifications are listed as follows and are highlighted in the revised manuscript.

In Page 1: “Haloperoxidase-like enzyme”→“Haloperoxidase-like enzymes”; “in a wide range of temperature”→“at a wide range of temperature”; “the present of phenol red”→“the presence of phenol red”; “resist Pseudomonas aeruginosa et al. bacteria.”→“resist typical marine bacteria like Pseudomonas aeruginosa.”; “even new types”→“even new types of”; “were applied to”→“are applied to”; “restricted”→“restricted to”;

In Page 2: “low long-term stability”→“short-term stability”; “as shown scheme 1”→“as shown in scheme 1”

In Page 3: “does not”→“did not”;

In Page 4: “PR”→“phenol red (PR)”; “and it’s used throughout the study”→“Thus, this doping composite was used throughout the study”; “the absorbance spectra were obtained in different reaction systems”→“the absorbance spectra of solutions were obtained in different reaction systems”.

In Page 6: “are calculated as shown in Table 1”→“calculated are shown in Table 1”;

In Page 9: “, and the medium”→“in the medium”; “absence of”→“the absence of”; “figure 5A right column”→“Figure 5A right column”; “reduced P. aeruginosa colonies”→“reduced number of P. aeruginosa colonies”. “Cell staining kit (K2081) stain”→“Cell staining kit (K2081)”.

In Page 10: “The reaction scheme is that”→“The reaction scheme was that”; “KBr, NaBr and NH4Br served as”→“KBr, NaBr and NH4Br were served as”; “Bacterial Adhesion Tests”→“Bacterial adhesion tests”.

In Page 11: “In the bacterial adhesion tests”→“For bacterial adhesion tests”; “Ti was dyed”→“Ti was stained”.

Round 2

Reviewer 2 Report

In overall, this manuscript was technically well revised. This revised manuscript meets the criteria of IJMS. Therefore, in my opinion, the revised manuscript can be accepted for publication.

Author Response

thank you very much for your efforts and comments.